# Effects of the Simulated Enhancement of Precipitation on the Phenology of *Nitraria tangutorum* under Extremely Dry and Wet Years

**DOI:** 10.3390/plants10071474

**Published:** 2021-07-19

**Authors:** Fang Bao, Zhiming Xin, Jiazhu Li, Minghu Liu, Yanli Cao, Qi Lu, Ying Gao, Bo Wu

**Affiliations:** 1Institute of Desertification Studies, Chinese Academy of Forestry, Beijing 100091, China; leejzids@caf.ac.cn (J.L.); caoyanli@caf.ac.cn (Y.C.); luqi@caf.ac.cn (Q.L.); yinggao@caf.ac.cn (Y.G.); 2Key Laboratory for Desert Ecosystem and Global Change, Chinese Academy of Forestry, Beijing 100091, China; 3Experimental Center of Desert Forestry, Chinese Academy of Forestry, Dengkou 015200, China; xzmlkn@163.com (Z.X.); slzxlmh@sina.com (M.L.); 4Inner Mongolia Dengkou Desert Ecosystem National Observation Research Station, Dengkou 015200, China

**Keywords:** plant phenology, desert plants, irrigation, extremely drought, extremely humidity

## Abstract

Plant phenology is the most sensitive biological indicator that responds to climate change. Many climate models predict that extreme precipitation events will occur frequently in the arid areas of northwest China in the future, with an increase in the quantity and unpredictability of rain. Future changes in precipitation will inevitably have a profound impact on plant phenology in arid areas. A recent study has shown that after the simulated enhancement of precipitation, the end time of the leaf unfolding period of *Nitraria tangutorum* advanced, and the end time of leaf senescence was delayed. Under extreme climatic conditions, such as extremely dry or wet years, it is unclear whether the influence of the simulated enhancement of precipitation on the phenology of *N. tangutorum* remains stable. To solve this problem, this study systematically analyzed the effects of the simulated enhancement of precipitation on the start, end and duration of four phenological events of *N. tangutorum*, including leaf budding, leaf unfolding, leaf senescence and leaf fall under extremely dry and wet conditions. The aim of this study was to clarify the similarities and differences of the effects of the simulated enhancement of precipitation on the start, end and duration of each phenological period of *N. tangutorum* in an extremely dry and an extremely wet year to reveal the regulatory effect of extremely dry and excessive amounts of precipitation on the phenology of *N. tangutorum*. (1) After the simulated enhancement of precipitation, the start and end times of the spring phenology (leaf budding and leaf unfolding) of *N. tangutorum* advanced during an extremely dry and an extremely wet year, but the duration of phenology was shortened during an extremely wet year and prolonged during an extremely drought-stricken year. The amplitude of variation increased with the increase in simulated precipitation. (2) After the simulated enhancement of precipitation, the start and end times of the phenology (leaf senescence and leaf fall) of *N. tangutorum* during the autumn advanced in an extremely wet year but was delayed during an extremely dry year, and the duration of phenology was prolonged in both extremely dry and wet years. The amplitude of variation increased with the increase in simulated precipitation. (3) The regulation mechanism of extremely dry or wet years on the spring phenology of *N. tangutorum* lay in the different degree of influence on the start and end times of leaf budding and leaf unfolding. However, the regulation mechanism of extremely dry or wet years on the autumn phenology of *N. tangutorum* lay in different reasons. Water stress caused by excessive water forced *N. tangutorum* to start its leaf senescence early during an extremely wet year. In contrast, the alleviation of drought stress after watering during the senescence of *N. tangutorum* caused a delay in the autumn phenology during an extremely dry year.

## 1. Introduction

Plant phenology has long been regarded as the most sensitive and accurate biological indicator to track climate change [1,2,3]. Therefore, mastering the response law of plant phenology to climate change is of substantial significance to predict the impact of future climate change on the processes of terrestrial ecosystems [4,5]. In an ecosystem with adequate amounts of water, plant phenology is primarily driven by temperature. For example, climate warming led to the advanced phenology of many woody plants in the spring and postponed phenology in the autumn in Europe [6,7,8], North America [9,10,11], China [12,13,14], the southern hemisphere [15,16] and other areas. However, in arid and semiarid ecosystems, the availability of water plays a key role in the regulation of plant phenology [17,18]. There have been relatively few studies to date on plant phenology in arid and semiarid areas in the world [19,20,21,22,23,24]. Chinese researchers have also focused on forests [12,13] and grassland ecosystems [25,26,27,28], while research on the phenology of desert plants has been extremely rare [18]. China’s desert ecosystem covers an area of approximately 1.65 million km^2^ and is a representative type of ecosystem in northwest China [29], with a fragile ecological environment and extreme sensitivity to climate change, particularly changes in precipitation [30,31]. *Nitraria tangutorum*, an endemic plant in desert and semi-desert regions, is a constructive and dominant species that controls the local landscape in northwestern China owing to its exceptional ability to fix sand and build sandbags that play a substantial ecological role in preventing or slowing down the movement of sand (Figure 1A). The leaf phenology of *N. tangutorum* is sensitive to water [18], and changes in its phenology help to track changes in local precipitation and affect the accumulation of carbon in this species and the local areas.

Many climate models predict that the precipitation in desert areas of northwest China will increase under future climate change conditions [32,33,34,35,36]. For example, the annual precipitation in Northwest China is projected to increase by 25 and 50% during the middle and end of the 21st century, respectively [32]. Based on an RCP8.5 scenario, the annual increase in precipitation during the end of the 21st century in some desert areas of northwest China will reach 100% or even higher than that at the end of the 20th century [36]. Previous studies have shown that short-term watering can promote the flowering and fruiting time of four short-lived plants in spring at the southern edge of the Gurbantunggut Desert [37]. Long-term watering can advance leaf unfolding and delay the time of leaf senescence in *N. tangutorum*; however, these effects were modified during extremely dry or wet years [18]. Thus, a key scientific issue in this study is to clarify how the effects of watering on the phenology of *N. tangutorum* are adjusted and to underscore the mechanisms during extremely dry or wet years in detail. To solve this issue, this study systematically observed an entire series of phenological events, including the beginning, end and duration, of four phenological periods of *N. tangutorum*, including leaf budding, leaf unfolding, leaf senescence and leaf fall, during years that were extremely dry and extremely wet with a simulation in the enhancement of precipitation. The aims of this study lay in (1) clarifying the similarities in the simulated enhancement of precipitation during the beginning, end and duration of each phenological period of *N. tangutorum* in extremely dry and wet years; (2) comparing the differences in the simulated enhancement of precipitation on the beginning, end and duration of each phenological period of *N. tangutorum* during extremely dry and wet years; and (3) revealing the mechanism by which extremely dry and wet years regulate the influence of the simulated enhancement of precipitation on the phenology of *N. tangutorum*.

## 2. Results

### 2.1. Interannual Dynamics of Meteorological Factors

The monthly average temperature and changes in daily precipitation during 2012 and 2013 are shown in Figure 2. The average annual temperature was 18.44 °C in 2012 and 20.44 °C in 2013. Compared with the local average annual temperature (7.6 °C), the values of CV (coefficient of variation) in 2012 and 2013 were +142.73 and +168.91%, respectively, which were “higher than the average level” and belong to “warm” years. There was no significant difference in the daily temperature within two years during the phenological observations (Figure 2A). Therefore, the effects of temperature could be ignored in this study when the effects of the simulated enhancement of precipitation on the phenology of *N. tangutorum* under extremely dry conditions and extreme amounts of precipitation were compared. The annual precipitation in 2012 (Figure 2B) and 2013 (Figure 2C) was 211.6 and 57.3 mm, respectively, and the CV values were +45.93 and −60.48%, respectively, compared with the local multi-year average precipitation (145 mm). Thus, 2012 could be classified as an extremely “wet” year, while 2013 could be classified as an extremely “dry” year. In the medium term, the frequency of small precipitation events (<3 mm) and heavy precipitation events (>15 mm) was the main reason for the difference in precipitation between the two years.

### 2.2. Effect of Simulated Enhancement of Precipitation on the Content of Soil Water

RMANOVA analysis showed that both the amount and time of the simulated enhancement of precipitation had significant effects on soil water content (*p* < 0.01), and their interaction was also significant (*p* < 0.01). The influence of simulated precipitation in the content of soil water in lower soil layers (0–10 and 10–20 cm) was the highest in spring, followed by autumn and summer in 2012 (Figure 3A,B). The seasonal variation of soil water content was relatively small in the 20–30 cm layer after the enhancement of the simulated precipitation in 2012. The order of influence degree among the treatments was 100% > 75% > 50% > 25% in most cases in 2012 (Figure 3A–C). However, the influence degree was the largest in the 75% treatments in most cases in the 0–10 and 10–20 cm soil layers in September in 2013 (Figure 3D,E). The order of influence degree among the treatments was 50% > 75% > 100% > 25% in most cases in the 20–30 cm soil layers in 2013 (Figure 3F).

### 2.3. Phenological Fluctuation during the Spring in Extremely Dry and Wet Years

A two-way ANOVA analysis demonstrated that interannual differences had significant effects on almost all the phenological events (with the exception of the beginning time of leaf unfolding) (*p* = 0.02), and the enhancement of precipitation had significant effects on the beginning times of leaf budding and leaf unfolding and the end time of leaf fall (*p* = 0.03). The interaction (Year∗Treatment) between interannual differences and the enhancement of precipitation only had observable effects at the end time of leaf fall (Table 1).

As shown in Figure 4, the beginning and end times of spring phenological events advanced after the simulated enhancement of precipitation whether the year was extremely dry or wet. The beginning time of leaf budding advanced 2.5, 1.5, 2.5 and 3.0 d, respectively, in +25, +50, +75 and +100% of irrigation treatment plots in 2012. The magnitudes of the relative changes were stronger in 2013, and the beginning time of leaf budding advanced 2.5, 4.0, 5.0 and 5.0 d, respectively. However, the influence of the simulated enhancement of precipitation during the relative change in spring phenological duration differed significantly owing to the difference in dry or wet years. After the simulated enhancement of precipitation, the duration of the leaf budding period shortened (+25% and +50% precipitation treatments) in a wet year (2012) and was prolonged in a dry year (2013) (Figure 4C) in contrast with the control. Similarly, after the simulated enhancement of precipitation, the duration of the leaf unfolding period was shortened in a wet year (2012) and prolonged in a dry year (2013) (Figure 4F) compared with that of the control. The specific values are shown in Table 2.

### 2.4. Phenological Fluctuation during the Autumn in Extremely Dry and Wet Years

It is apparent from Figure 4 that the impact of the simulated enhancement of precipitation on phenology during the autumn differed notably from that in the spring. After the simulated enhancement of precipitation, the beginning times of leaf senescence and leaf fall start advanced during the extremely wet year but were delayed during the extremely dry year (Figure 5A–E). As a result, the durations of leaf senescence and leaf fall phenology were prolonged after the simulated enhancement of precipitation compared with the control (Figure 5C,F). The specific values are shown in Table 2.

### 2.5. The Correlation between Phenology and the Simulated Increase in Precipitation

A simple linear regression analysis showed that the beginning times of leaf budding, leaf unfolding and leaf senescence had a marginally significant linear negative correlation with the increase in simulated precipitation during an extremely wet year (0.05 < *p* ≤ 0.10), but this correlation was not significant during an extremely dry year (Figure 6A).

All the phenological events, including leaf budding, leaf unfolding, leaf senescence and the end time of leaf fall, had a significantly linear correlation with the simulated increase in precipitation in the year that was extremely dry (*p* < 0.05) (Figure 6B). Among them, the end time of leaf budding displayed a significant linear negative correlation with the simulated increase in precipitation. However, there was a significant linear positive correlation between the end times of leaf unfolding, leaf senescence, leaf fall and the simulated increase in precipitation (*p* < 0.05) (Figure 6B). During an extremely wet year, the end times of leaf budding and leaf fall had a marginally significant linear negative correlation with the simulated increase in precipitation (0.05 < *p* ≤ 0.10) (Figure 6B).

There was a significant positive correlation between the duration of the leaf budding period and the simulated increase in precipitation during an extremely dry year (*p* < 0.05), but this correlation was not significant during an extremely wet year (Figure 6C). There was a significant positive correlation between the duration of the leaf unfolding period and the increase in simulated precipitation during an extremely dry year (*p* < 0.05) and a negative correlation during an extremely wet year, but the correlation was not significant (Figure 6C). There was a significant positive correlation between the duration of leaf senescence and the simulated increase in precipitation in both an extremely dry and an extremely wet year (*p* < 0.05) (Figure 6C). There was a significant positive correlation between the duration of the leaf fall period and the simulated increase in precipitation during an extremely dry year (*p* < 0.05), but there was no significant correlation during an extremely wet year (Figure 6C).)

## 3. Discussion

### 3.1. Effect of the Simulated Enhancement of Precipitation on Phenology in the Spring

Bao et al. (2020) averaged the data for seven consecutive years from 2012 to 2018 and found that the simulated enhancement of precipitation could advance the spring phenology (leaf unfolding period) of *N. tangutorum*. On this basis, this study analyzed the influence of the simulated enhancement of precipitation in phenology (leaf budding) in early spring in more detail and found that under the conditions of a simulated enhancement of precipitation, whether in extremely wet or dry years, the start and end times of the leaf budding and leaf unfolding of *N. tangutorum* advanced. The results were closely related to the supplementation of soil moisture by the long-term simulated enhancement of precipitation [18,38]. Most of the precipitation in this study area was concentrated from June to September, and the simulated precipitation was enhanced in mid-May, which corresponded to the middle and late stages of the leaf unfolding period of *N. tangutorum*. Thus, the leaf budding and unfolding stages of *N. tangutorum* were dependent on soil moisture [39]. Although the phenological start and end times of *N. tangutorum* during early spring and spring had the same change trend (advanced) when subjected to the simulated enhancement of precipitation in extremely dry and wet years, the simulated enhancement of precipitation had a different influence on the start and end times. This difference led directly to opposite changes in the duration of the leaf budding and leaf unfolding periods in different years. In an extremely dry year, the simulated precipitation had a stronger influence on the start time of leaf budding and leaf unfolding than the end time, which directly led to the extension of the duration of the leaf budding and leaf unfolding of *N. tangutorum* following the simulated precipitation treatment. In extremely wet years, the simulated enhancement of precipitation had an opposite influence during the beginning and end times of *N. tangutorum*’s leaf budding and leaf unfolding periods, i.e., it had less of an influence on the start times of the leaf budding and leaf unfolding periods than the end times, thereby resulting in a shortening of the duration of *N. tangutorum*’s leaf budding and leaf unfolding periods. The regulation of years of extreme drought and precipitation on the simulated enhancement of precipitation and spring phenology lay in the regulation of the influence degree of its start and end time.

### 3.2. Effects of the Simulated Enhancement of Precipitation on Autumn Phenology

The onset of autumn phenology (leaf senescence and leaf fall) is precisely controlled by a series of biochemical processes, such as the degradation of nutrients and macromolecules [40]. Precipitation and the precipitation pattern [41,42], temperature, photoperiod [40] and even the time of the occurrence of spring phenology [43,44] can influence the interannual variation in autumn phenology. This study confirmed that the results were exceptional under extremely wet conditions. The start and end times of leaf senescence and leaf fall of the autumn phenology of *N. tangutorum* advanced instead of being delayed during extremely wet years after the simulated enhancement of precipitation. This could be related to the water stress of *N. tangutorum* in extremely wet years, and excessive water could force *N. tangutorum* to start its autumn phenology early [41,45]. In extremely wet years, the simulated enhancement of precipitation had a greater influence in the beginning time of autumn phenology than on the end time, which caused the duration of the leaf senescence and leaf fall periods of *N. tangutorum* to be prolonged, even in extremely wet years. In contrast, in extremely dry years, the start and end times of the leaf senescence and leaf fall of *N. tangutorum* were delayed after the treatment of the simulated enhancement of precipitation. The simulated precipitation had a weaker influence on the beginning times of leaf senescence and leaf fall than the end times, which also resulted in the extension of the duration of leaf senescence and leaf fall after treatment with simulated precipitation. The results correlated with the improvement in activities of the photosynthesis-related enzymes of *N. tangutorum* [29,46] and a decrease in the degradation of chlorophyll during the senescence of *N. tangutorum* leaves [18,40] under extremely dry conditions. The regulation of extremely dry and wet years on the simulated enhancement of precipitation on the autumn phenology of *N. tangutorum* lay in their different influencing mechanisms with a negative effect in wet years and a positive effect in dry years.

## 4. Materials and Methods

### 4.1. Sample Area Description

This study was conducted at the Desert Ecosystem Irrigation Platform (106°43′ E, 40°24′ N, 1050 m), which was set up in Dengkou County, Inner Mongolia Autonomous Region, China, in 2008. The local climate type is a mid-temperate continental monsoon, with an average annual temperature of 7.6 °C (1978–2007), an average annual precipitation of 145 mm (1978–2007) and an average annual evaporation of 2381 mm. *N. tangutorum* is the dominant plant in the research site, and it often forms clumps that are approximately 1~3 m high and 6~10 m in diameter with vegetation coverage of approximately 45~75% [29]. *N. tangutorum* sandbags are distributed in patches on the surface of hard mud, and other plants, such as *Psammoch loavillosa* and *Artemisia ordosica*, are occasionally found around the clumps. The soil type on *N. tangutorum* clumps is aeolian sand, and the open space between clumps is grey-brown desert soil.

### 4.2. Simulated Enhancement in Precipitation

Five precipitation enhancement gradients were established on the simulated enhancement of the precipitation platform, including the control (natural precipitation), which was used as the baseline for the enhancement of precipitation by 25% (+25%), 50% (+50%), 75% (+75%) and 100% (+100%) of the local average annual precipitation (145 mm). Each treatment had four replicates, including 20 sample plots with intervals of 5~15 m, which were arranged in random blocks. The water source for the enhancement of precipitation originated from a well near the sample plot, and the well water was pumped to a pressure water tank for temporary storage. After diversion by the water separator, the water in the water tank was delivered to the full-automatic sprinkler irrigation system (consisting of a base and two rotating arms with a length of 6 m) installed above each sample plot (sandbag) through water pipes. Driven by the water flow, the sprinkler irrigation system rotated at a constant speed with both arms, and the water was simultaneously sprayed from the rotating arms at multiple points to ensure that the sample plot received a uniform amount of water (Figure 1B). The enhancement of precipitation was controlled using a water meter. During the growing season from May to September in 2012 and 2013, the precipitation was enhanced once a month. The increases in precipitation for the control, 25, 50, 75 and 100% treatments were 0, 7.25, 14.5, 21.75 and 29 mm, respectively. More detailed information on the experimental design and the irrigation system can be found in Bao et al., 2017 and Bao et al., 2020.

### 4.3. Phenological Observations

The frequency of phenological observation was 2 days and involved recording the time of occurrence of a series of phenological events, such as the beginning and end times of leaf budding, leaf unloading, leaf senescence and leaf fall (Table 3). The phenological events were observed based on Chinese standard regulations for the observation of plant phenology [47].

In this study, soil samples from soil layers that were 0–10, 10–20 and 20–30 cm deep were collected to analyze the influence of simulated precipitation on the content of soil water (mass), which was determined by drying. The sampling time was 1 d before each treatment of enhanced precipitation and 1, 3 and 7 d after each treatment in May, July and September in 2012, and only September in 2013. Meteorological data, such as natural precipitation and temperature, originated from the standard meteorological observation station near the sample plot.

### 4.4. Statistical Analysis

Before phenological data processing, the date at which phenological events occurred was converted into days. The duration of a phenophase is the difference between the end and beginning times. The influence of the enhancement of precipitation on phenological events was determined by calculating the difference between the time of the occurrence of phenological events in the sample plot and the time of the occurrence of phenological events in the control plot. If the number of days was reduced compared with the control, it was deemed advanced. Otherwise, it was postponed. Repeated measurement ANOVA (RMANOVA) analysis was used to test the effects of enhanced precipitation on soil water content at depths of 0–10, 10–20 and 20–30 cm, respectively. Two-way ANOVA analysis was utilized to analyze the effects of the enhancement of precipitation, interannual variation and their interaction on the phenology of *N. tangutorum*. The significance level was set at α = 0.05. The correlation between simulated precipitation and phenology was analyzed using a simple linear regression method. All statistical analyses were completed in SPSS 20.0 (IBM, Inc., Armonk, NY, USA), and Microsoft Excel 2019 (Redmond, WA, USA) was adopted for plotting.

## 5. Conclusions

Climate change affects plant phenology, which, in turn, influences ecosystem functions, such as carbon and nitrogen cycles and the interaction of species, among others, and reacts to climate change [39,40]. This study found that whether the years were extremely dry or wet, the simulated enhancement of precipitation had a greater impact on the phenology of *N. tangutorum* in the autumn than in the spring, which was consistent with previous studies [18]. The novelty of this study was its systematic analysis of the influence of the simulated enhancement of precipitation during the whole process (start, end and duration) of a phenological event of *N. tangutorum* under extremely dry and wet conditions. This study revealed that there were seasonal differences in the phenology of *N. tangutorum* following the regulation of the stimulated enhancement of precipitation in response to extremely dry or wet years. The effect of extreme climate on spring phenology primarily originated from the difference in influence degree during the beginning and end time of the leaf budding and leaf unfolding periods. The influence on autumn phenology was primarily owing to the biochemical mechanisms that promoted the onset of autumn phenology.

## Figures and Tables

**Figure 1 plants-10-01474-f001:**
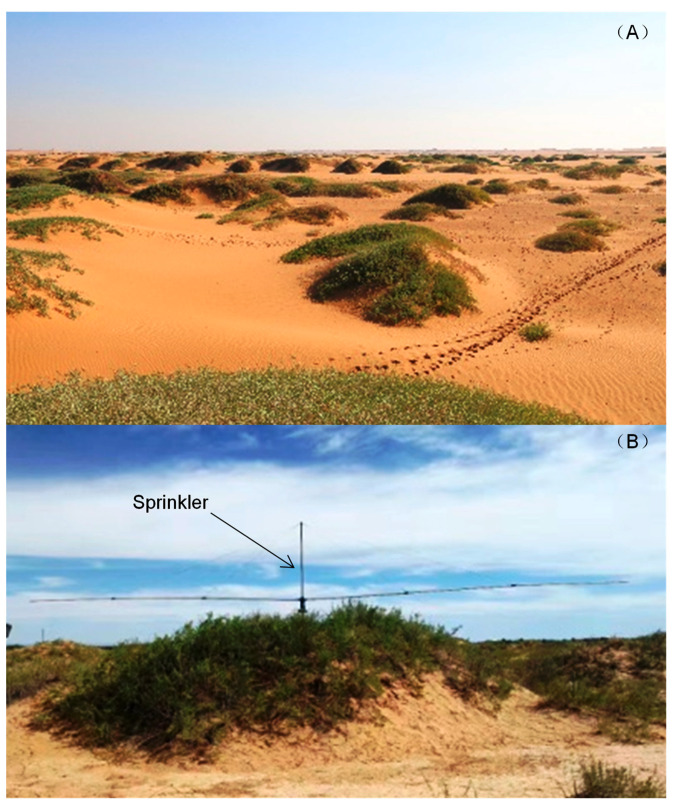
Nitraria tangutorum desert ecosystem of the study site (**A**) and sprinkler on sandbags ((**B**) [18]) in Inner Mongolia, China.

**Figure 2 plants-10-01474-f002:**
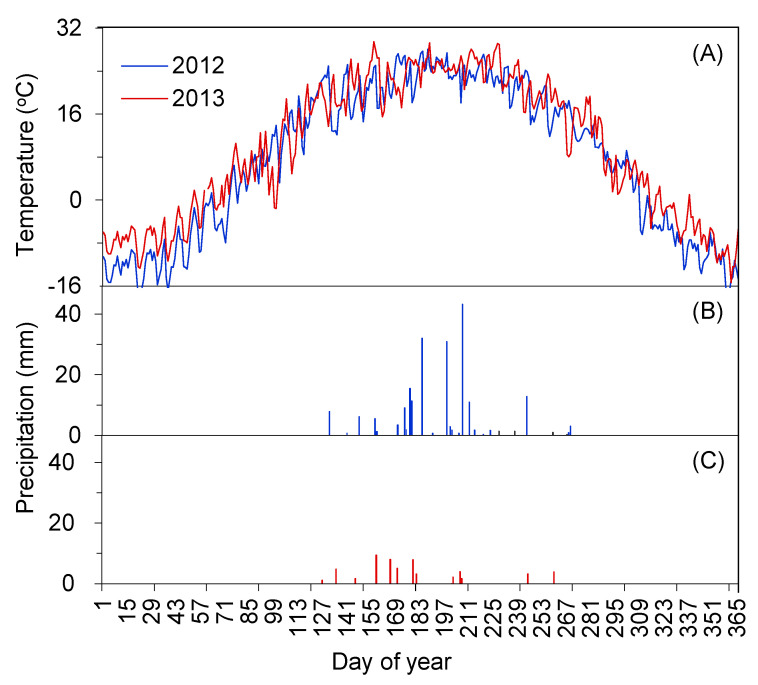
Dynamic changes of daily temperature (**A**), daily precipitation in 2012 (**B**) and daily precipitation in 2013 (**C**). Solid arrows indicate the times of the simulated enhancement of precipitation.

**Figure 3 plants-10-01474-f003:**
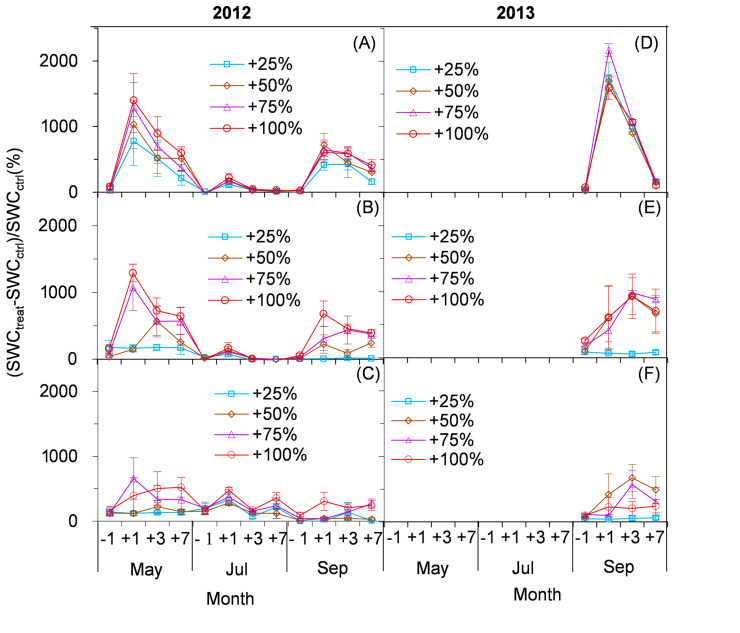
Change trend of soil water content (SWC) at 0–10 (**A**,**D**), 10–20 (**B**,**E**) and 20–30 cm (**C**,**F**) one day before (−1) and one day (+1), three days (+3) and seven days (+7) after the enhancement of precipitation compared with the control in 2012 and 2013.

**Figure 4 plants-10-01474-f004:**
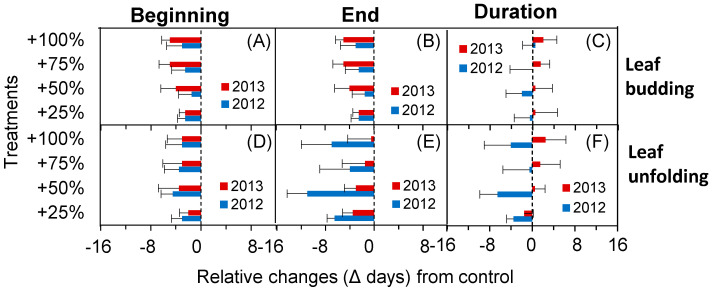
Effect of the simulated enhancement of precipitation on the time of the beginning (**A**,**D**), end (**B**,**E**), and duration (**C**,**F**) of the spring phenological events.

**Figure 5 plants-10-01474-f005:**
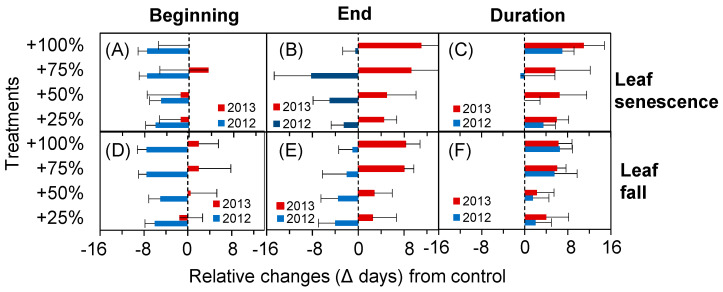
Effect of the simulated enhancement of precipitation on the time of the beginning (**A**,**D**), end (**B**,**E**), and duration (**C**,**F**) of the autumn phenological events.

**Figure 6 plants-10-01474-f006:**
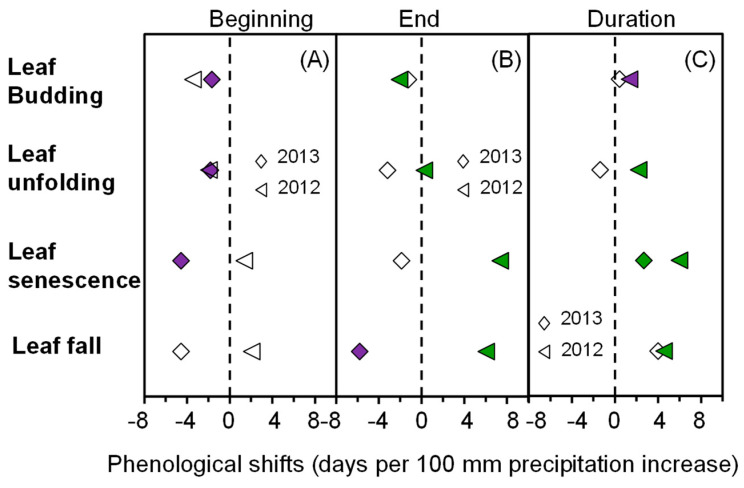
Slope of the linear regression line between phenological events, Beginning (**A**), End (**B**), Duration (**C**), and the simulated increase in precipitation. Green indicates a significant correlation (*p* ≤ 0.05), and purple indicates a close significant correlation (0.05 < *p* ≤ 0.10). A blank indicates the lack of a significant correlation.

**Table 1 plants-10-01474-t001:** The *p* values of a two-way ANOVA of the influence of precipitation enhancement treatment and its interaction on the phenology of *Nitraria tangutorum*.

	Leaf Budding			Leaf Unfolding			Leaf Senescence			Leaf Fall		
	Start	End	Duration	Start	End	Duration	Start	End	Duration	Start	End	Duration
Year	<0.01	<0.01	<0.01	0.37	0.06	<0.01	<0.01	<0.01	<0.01	<0.01	<0.01	<0.01
Treatment	<0.01	0.11	0.70	0.02	0.12	0.84	0.33	0.42	0.17	0.10	0.03	0.18
Year∗Treatment	0.97	0.97	0.93	0.59	0.77	0.29	0.06	0.50	0.74	0.97	0.01	0.89

**Table 2 plants-10-01474-t002:** Duration of each phenological period under different treatment conditions ^1^ in 2012 and 2013 (means ± SE).

Year	Treatments	Spring Phenology		Autumn Phenology	
		Leaf Budding	Leaf Unfolding	Leaf Senescence	Leaf Fall
2012	Control	24.67 ± 1.26	20.50 ± 3.30	80.50 ± 0.96	88.50 ± 1.26
	+25%	21.00 ± 1.73	17.00 ± 3.11	84.00 ± 1.63	90.50 ± 1.71
	+50%	19.50 ± 2.50	14.00 ± 0.82	80.50 ± 3.77	90.00 ± 3.16
	+75%	21.50 ± 3.10	20.00 ± 3.56	79.75 ± 7.17	94.00 ± 3.92
	+100%	22.00 ± 2.83	16.50 ± 2.63	87.50 ± 3.20	95.00 ± 2.89
2013	Control	21.50 ± 0.96	16.00 ± 1.41	76.50 ± 2.75	79.75 ± 4.73
	+25%	22.00 ± 0.82	14.50 ± 0.50	82.50 ± 1.71	83.75 ± 1.18
	+50%	22.00 ± 2.45	16.50 ± 1.26	83.00 ± 1.29	82.00 ± 1.83
	+75%	23.00 ± 2.38	17.50 ± 1.71	82.25 ± 2.66	85.75 ± 3.64
	+100%	23.50 ± 2.06	18.50 ± 2.63	**87.50 ± 5.74**	86.00 ± 2.92

^1^ indicates a significant difference compared with that of the control. Values in bold are significantly different from control (*p* < 0.05).

**Table 3 plants-10-01474-t003:** Phenological observation and observation methods of *Nitraria tangutorum*.

Phenological Events		Observation Methods
Leaf budding	Beginning period	In all leaf buds on the whole sandbag, as long as the leaf budding was observed, the beginning of leaf budding period was recorded.
	End period	Most of the leaf buds on the whole sandbag had split, and the tips of leaflets were clearly visible.
Leaf unfolding	Beginning period	The beginning of leaf unloading was recorded as long as at least one young leaf had completely extended and spread out completely from one or more leaf buds had been observed on the whole sandbag.
	End period	More than 90% of the young leaves on leaf buds of the whole sandbag have been completely spread, which is recorded as the end period of leaf unfolding.
Leaf senescence	Beginning period	Approximately 5% of all the leaves on the whole sandbag have begun to turn yellow, which was recorded as the beginning of leaf senescence.
	End period	More than 90% of the leaves on the whole sandbag have turned yellow, which was recorded as the end period of leaf senescence.
Leaf fall	Beginning period	When approximately 5% of the leaves on the whole sandbag fell off, the beginning of leaf fall was recorded.
	End period	When more than 90% of the leaves on the whole sandbag had fallen off, the end stage of leaf fall was recorded.

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
