# Peer review of "Effects of the Simulated Enhancement of Precipitation on the Phenology of Nitraria tangutorum under Extremely Dry and Wet Years"

_plants, 2021, doi:10.3390/plants10071474_

Round 1
Reviewer 1 Report
In the introduction is not described Nitraria tangutorum, the importance of this species and the potential phenology change impact on the ecosystem.
The aim of the work is not sufficiently justified
Materials and Methods are not clear.
Ex Is not clear why the first year (incorrectly indicated 2008) they applicate 10 supplementary irrigation and the second year (incorrectly indicated 2009) 5.
Is not indicated how man years were considered to determined climatic parameters. (average annual temperature are very different from average level).
Soil water content and the figure (2) was referred only to one year and is not indicated the methodology for its determination.
Some statistical measures as standard deviation were not correctly expressed.
Author Response
Point 1: In the introduction is not described Nitraria tangutorum, the importance of this species and the potential phenology change impact on the ecosystem.
Response point 1: Thanks for your comments. The importance of Nitraria tangutorum has been described, and the potential phenology change impact on the ecosystem has also been added to the revised manuscript(L66-73). Please see details in the revised manuscript.
Point 2: The aim of the work is not sufficiently justified
Response to point 2: Thanks for your comments. The aim of the the manuscript has been reconstructed and rewritten, please see details in the revised manuscript(L93-100).
Point 3: Materials and Methods are not clear. Ex Is not clear why the first year (incorrectly indicated 2008) they applicate 10 supplementary irrigation and the second year (incorrectly indicated 2009) 5.
Response to point 3: Thanks for your comments. The irrigation plat form was set in 2008 (This is correct, and the phenological observations began four years later in 2012), we did applicate ten supplementary irrigations in the first year. However, considering the poor effect of low enhancement of precipitation during each irrigation, the precipitation was increased once a and five times(May to September) a year since 2009. We added the explanation to the revised manuscript (L306-311). A photograph of sprinkler on N. tangutorum sand bag has also been added to the revised manuscript (Figure6). Please see details in the revised manuscript.
Point 4: Is not indicated how many years were considered to determined climatic parameters. (average annual temperature are very different from average level).
Response to point 4: Thanks for your comments. The specific long-term (1978-2007) period has been shown in the revised manuscript (L109,292). Please see details in the revised manuscript.
Point 5: Soil water content and the figure (2) was referred only to one year and is not indicated the methodology for its determination.
Response to point 5: Thanks for your comments. I apologize for the inconvenience caused by this problem. More detail information in Soil water content determination has been added to the revised manuscript (L327-331), and the results have been rewritten correspondingly. The Figure 2 has been replotted with new data both from 2012 and 2013. See details in the revised manuscript please.
Point 6: Some statistical measures as standard deviation were not correctly expressed.
Response to point 6: I really appreciate your keen eye and spot the standard deviation error in Figure 2. This Figure has been replotted and see details in the revised manuscript please.
The language has been polished by a native speaker. Please see the attachment for the Certification.

Reviewer 2 Report
This is an interesting paper trying to decipher the effect of future climate changes on the phenology of a typical desert plant by means of a simulated rain, superimposed on natural precipitation during two consecutive years – one more rainy, the next with scarce precipitation compared to the norm. The work deserves publication after a minor revision. Below are listed mainly technical remarks:
Title – “under extreme drought and wet years” – better sounds drought and humidity years, or dry and wet years, but there are some references using exactly this expression, for example “Larson, M. M., Kost, D. A., & Vimmerstedt, J. P. (1995). Establishment of trees on minesoils during drought and wet years. International Journal of Surface Mining and Reclamation, 9(3), 99-103.”
Abstract - line 14 “with increased precipitation and uncertainty” – not clear enough. Did the authors mean increased rain quantity and unpredictability?
Line 19 – influence of … on the phenology
Line 25 – in order to reveal
Line 37 – “the regulation of autumn phenology lay in different influencing mechanism” – should be specified more in details
Lines 78-79 - the aim “(2) revealing how to regulate the influence of simulated enhancement of precipitation on the phenology of N. tangutorum in extremely dry and wet years” – could be better explained
Line 135 “budding period” but in fig. 3F the legend is about leaf unfolding
Line143-144 –“ the leaf senescence and leaf fall start times advanced in the extremely wet years but were delayed in the extremely dry years” – plural but in fact the observations are during one wet and one dry year.
Line 186 - Figure 5 symbols – red and blue colors are used for year 2013 and 2012, other colors could be used here to avoid ambiguity
Author Response
Point 1: Title – “under extreme drought and wet years” – better sounds drought and humidity years, or dry and wet years, but there are some references using exactly this expression, for example “Larson, M. M., Kost, D. A., & Vimmerstedt, J. P. (1995). Establishment of trees on mine soils during drought and wet years. International Journal of Surface Mining and Reclamation, 9(3), 99-103.”
Response to point 1: Thanks for your suggestion. The word drought has been replaced with the word dry in the revised manuscript.
Point 2: Abstract - line 14 “with increased precipitation and uncertainty” – not clear enough. Did the authors mean increased rain quantity and unpredictability?
Response to point 2: Thanks for your kindly reminding. We have rewritten this sentence in the revised manuscript according to your suggestion.
Point 3: Line 19 – influence of … on the phenology
Response to point 3: Thanks for your editing. The error has been fixed in the revised manuscript according to your suggestion.
Point 4: Line 25 – in order to reveal
Response to point 4: Thanks for your patiently editing. The sentence has been edited by a native English speaker, please see details in the revised manuscript.
Point 5: Line 37 – “the regulation of autumn phenology lay in different influencing mechanism” – should be specified more in details
Response to point 5: Thanks for your suggestion. It has been specified more in details in the revised manuscript(L36-43).
Point 6: Lines 78-79 - the aim “(2) revealing how to regulate the influence of simulated enhancement of precipitation on the phenology of N. tangutorum in extremely dry and wet years” – could be better explained
Response to point 6: Thanks for your suggestion. The aim part in the introduction has been reconstructed and rewritten, see details in the revised manuscript please.
Point 7: Line 135 “budding period” but in fig. 3F the legend is about leaf unfolding
Response to point 7: Thanks for your carefulness. “budding period” here should be leaf unfolding. The error has been fixed in the revised manuscript.
Point 8: Line143-144 –“the leaf senescence and leaf fall start times advanced in the extremely wet years but were delayed in the extremely dry years” – plural but in fact the observations are during one wet and one dry year.
Response to point 8: Thanks for your editing. The plural errors have been fixed in the revised manuscript according to your suggestion.
Point 9: Line 186 - Figure 5 symbols – red and blue colors are used for year 2013 and 2012, other colors could be used here to avoid ambiguity
Response to point 9: Thanks for reminding. The Figure 5 has been replotted in the revised manuscript according to your suggestion.
Round 2
Reviewer 1 Report
Line 40-43: this parameters (photosynthesis, chlorophyll,etc.) were not observed in this paper.
Line 45: In the keyword is better to use different words respect to the words of the title.
Line 54: delete the citations
Line 82: which mechanism you underscored ?
Line92-94: as above. which mechanism you revealed?
Line 97-98: in fig 1 are represented 11 montly average temperature
Line 99: average annual temperature is referred as precipitation (1978-2007) line 109 ?
Line 100-101: standard deviation is not a %.. Is not clear what you indicate. (rewrite)
Line 108 : as above
Line 119: insert this information in M &M in full way
Line 120-122: were are indicated this results?
Line 122-124: during the experiment (2012 2013) do you change time of simulated precipitations? not clear rewrite or delete.
Line 129-132: you have done ANOVA. this differences are significative?
Line 144: leaf unfolding is 0.02 end time of leaf fall is 0.03.
Line 151-153: please insert data or days of the year of leaf budding
Line 224-228: You indicate that simulated precipitation was enhanced to the middle and late stage of leaf budding. So the treatments can’t influences leaf budding start/beginning
Materials & methods: are not well write there are unnecessary information and sometimes repeated
Line 272 -279: I suggest to describe the experimental site position briefly and in a concise way.
292 293 you just indicate average annual precipitation in line 281
Line 303 – 308: I suggest describing experimental conditions only for paper experimental activity (2012 and 2013)
The irrigation scheduling is empiric and start form May to September (growing season) with one irrigation for months. Do you irrigate the same date for 2012 and 2013? In fig 2 are reported only 3 irrigations. How do you regulate respect to natural precipitations?
I want to know The data of first supplementary irrigation and the data of beginning leaf budding.
Line 327-331: in the results you wrote that you have done anova analysis (line 119) you must indicate experimental design (exp. N of replicate etc) and analysis in M &M
Fig. 1 (A) the monthly average temperature bars are 11 and not 12.
Fig 2 data of 2013 are incomplete: I suggest to indicate in M & M that the data were registered only in a briefly period for … reason
In the fig are indicated the time of simulated precipitations but are only 3. In M&M and fig 1 you indicate 5
And also the natural precipitations precipitation are not correctly indicated.
May be if you will use the same legend on the fig 1 and 2 (days of the year or date) the paper will be more legible.
